# Impacts of Livestock Production on Grassland Grasshopper Disasters

**Sibo Liu** [1,2,3], **Meimei Chen** [1], **Ping Li** [1,*], **Taogetao Baoyin** [2], **Xiangyang Hou** [1,4] **and Guomei Yin** [3]

[1] Institute of Grassland Research, Chinese Academy of Agricultural Sciences, Hohhot 010010, China; lsbss258@126.com (S.L.); chenmeimei@caas.cn (M.C.); houxiangyang@caas.cn (X.H.)

[2] College of Ecology and Environment, Inner Mongolia University, Hohhot 010021, China; bytgt@imu.edu.cn

[3] Inner Mongolia Academy of Agricultural and Animal Husbandry Sciences, Hohhot 010031, China; gmynmg@126.com

[4] College of Grassland Science, Shanxi Agricultural University, Jinzhong 030801, China

\* Correspondence: liping01@caas.cn; Tel.: +86-152-4715-4700

**Abstract:** Grasshopper disasters threaten grassland animal husbandry, and overgrazing is widely recognized as one of the main causes of locust infestation in grassland regions. However, the impact of overgrazing on grasshopper disasters remains unclear. To address this knowledge gap, this study interviewed 541 households living in locust-prone areas in Inner Mongolia, China. The generalized Poisson model and OLS regression examined the relationship between herders' production behavior and locust disasters. The results showed that 42% of the herders had suffered from locusts more than three times over the past 15 years, with an average of 49 ha of grassland damaged per household. In addition, with the increase in grazing rates, the scale of locust disasters decreased before it increased. The results also showed that operating grassland areas and feeding forage reduced locust disasters significantly, while renting grassland areas and grazing rates worked oppositely. These results suggest that grazing intensity can make a significant difference in the occurrence of locust disasters.

**Keywords:** grassland grasshopper disasters; herders; livestock production; overgrazing; Inner Mongolia

## 1. Introduction

### 1.1. Background

Grasshopper disasters are a major global biological concern that endangers the security of agriculture and animal husbandry. Locust plagues occur frequently and cause damage to human lives in space and time [1–4]. In the event of such a disaster, locusts can completely devour farmlands and rangelands, leading to crop failure, food shortage, famine, and even deaths [5]. The most recent locust infestation in 2020 destroyed more than 70,000 hectares of land in Somalia and Ethiopia, putting millions of lives at unprecedented risk [3,4,6].

Locusts can cause enormous losses to grassland livestock husbandry and grassland ecosystems by eating large amounts of vegetation in a short period of time [7,8]. The damage to grassland caused by the locust plague is not only devastating but difficult to recover within 20 years, if not irreversible. The consequent economic losses can persist. The ecological losses caused by locust infestation, such as desertification and degradation of grassland, are incalculable [9]. Moreover, locust outbreaks decrease the income of local herders and threaten the sustainability of their livelihoods and welfare [6]. In 2003, locusts overran more than 15 million hectares of grasslands in North China, causing panic in local cities. Given their perniciousness, both grassland decision-makers and local citizens are eager to keep themselves updated on the regulations and key drivers of locust outbreaks.

Many factors, such as long-term drought, global warming, extreme weather events, and changes in land use and land cover, can influence locust disasters [10,11]. In the case of the African desert locust outbreaks reported by the FAO, six large infestations between 1912 and 1989 were associated with warming-drying climate conditions, and the 2020

disaster was thought to have been caused by global warming [12,13]. Eco-environmental deterioration is also considered a significant driver of locust outbreaks [14]. In addition, the bare land caused by locust outbreaks is a favorable environment for locust spawning and reproduction, leading to recurrent, large-scale grassland locust outbreaks [15].

Grazing, as the main use of natural grasslands, can affect communities of both plants and insect herbivores. In particular, overgrazing by domestic animals often results in vegetation degradation and soil erosion [16]. Previous studies helped build a knowledge map of locust outbreaks and control on grasslands [17,18]. Díaz et al. [19] pointed out that grazing was the main influence on promoting the grassland and exerted far-reaching effects on plant traits such as promoting the growth of annual over perennial plants, short plants over tall plants, and procumbent plant architectures. Take the outbreak of the Mongolian (*Oedaleus decorus asiaticus*) and Asian (*Locusta migratoria migratoria*) locust outbreaks. Cease [20] reported that heavy grazing promoted the outbreaks of locusts by reducing the plant nitrogen content, while Peng et al. [21] found that enhancing the grassland vegetation coverage up to 50% was the most fundamental method for reducing the frequency of locust disasters. Karpakakunjaram et al. [22] also pointed out that reasonable agricultural activities such as rotational grazing, reasonable mowing, and scientific management could regulate and reduce disasters, but grassland degradation caused by unreasonable overgrazing, excessive reclamation, and mining may contribute to disasters. Decreased vegetation coverage, grassland degradation, and desertification due to overgrazing would increase bare lands that are ideal habitats for grasshoppers [23]. Therefore, it is believed that the livestock production behavior of herders affected the outbreak of grassland grasshopper disasters.

Although past studies have demonstrated that overgrazing can affect grasshopper disasters, few researchers have explored the relationships between production behaviors and grasshopper disasters from the perspective of herders' livestock production. It remains unclear whether and how herders' activities in livestock management influence the frequency and severity of grasshopper disasters. Finding a good answer to the question of how herders' production behavior, especially overgrazing, contributes to grasshopper disasters will facilitate the prevention and control of grasshopper disasters at the household level.

Inner Mongolia's grassland, as part of the Eurasian steppe, is an essential livestock base and has been plagued by grasshoppers during its long nomadic history [24]. The region is not only representative in terms of grassland ecology, livestock production, and pastoral society, but also one of the most severely overgrazed grassland areas in China. Jin et al. [25] investigated more than 1700 households in Inner Mongolia and found that 37% of them engaged in serious overgrazing. This empirical study was intended to study the impact of human activities on grasshopper disasters. Based on the data from interviews with herders in grasshopper disaster-prone areas of Inner Mongolia, the generalized Poisson model was used to explore the impact of herders' production behavior, especially overgrazing, on grasshopper disasters at the microscopic level of herders. The results of the study are expected to provide references for grasshopper disaster control in arid and semi-arid grasslands around the world.

### 1.2. Literature Review

Overgrazing is the result of irrational production behaviors in grassland animal husbandry [26]. Long-term overgrazing can have a negative impact on grassland vegetation and soil, which in turn can lead to changes in locust communities [24,27]. On the one hand, the intensity of grazing, rotational or seasonal aspects of the grazing regime, and other grassland management practices have an impact on characteristics of grasslands such as vegetation height, biomass, and plant species. In turn, these factors can influence the oviposition, dispersal and feeding behaviors of grasshoppers, thereby affecting the dynamics within grasshopper communities [28,29]. On the other hand, overgrazing will lead to soil quality decline, soil moisture reduction, limit the growth of plants, and even exacerbate soil erosion, resulting in drought, desertification, and other problems [10,30]. In addition, overgrazing will lead to the disorderly consumption of grassland plants, causing

the rapid loss of nutrients in the soil, such as erosion and the loss of soil nitrogen that have degraded grasslands toward low-nitrogen plants [14]. Grassland degradation expands the exposed area of the soil, reduces the resources of beneficial wildlife and insect predators, and provides more space for the spawning and breeding of grasshoppers in some habitually arid habitats [15,31,32]. Therefore, our hypothesis is that livestock production behaviors resulting in overgrazing have a positive impact on locust disasters. The mechanism by which animal husbandry production behavior impacts grassland locust disasters is shown in Figure 1.

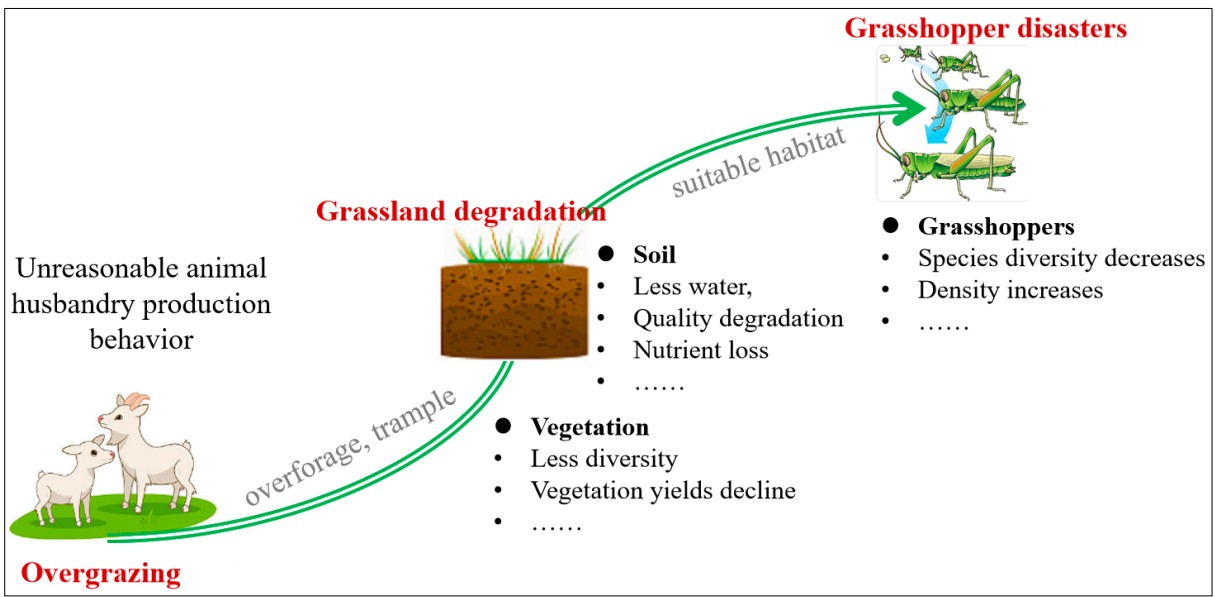

**Figure 1.** The mechanism by which animal husbandry production behavior influences grassland grasshopper disasters.

## 2. Materials and Methods

### 2.1. Study Area

Inner Mongolia is in the north of China, with the longitude and latitude at 97°12′ E~126°04′ E and 37°24′ N~53°23′ N, respectively (Figure 2). Here, there are 54.33 million hectares of grasslands. There is a temperate continental climate, with an average annual temperature of 5 °C, which increases gradually from east to west. The annual precipitation is 318 mm and declines from east to west. In line with rainfalls, grassland types range from meadow steppe, typical steppe, and sandy steppe to desert steppe from east to west [33]. Forage yield decreases from east to west as well, with the average productivity ranging from more than 1870 kg/ha in the east to less than 330 kg/ha in the west.

In Inner Mongolia, grasshopper species that can cause serious damage include *Oedaleus decorus asiaticus* Bei–Bienko, *Dasyhippus barbipes* (Fischer-Waldheim), *Bryodema luctuosum* (Stoll), *Pararcyptera microptera meridionalis* (Ikonnikov), *Bryodemella tuberculatum dilutum* (Stoll), *Bryodemella holdereri holdereri* (Krauss), and others (see Appendix A Table A1) [34]. The dominating species are *Oedaleus decorus asiaticus* Bei–Bienko and *Dasyhippus barbipes* (Fischer-Waldheim), due to their wide adaptability, diet, high reproductive capacity, and migration ability, which allow them to spread across different types of grassland areas [35]. Grasshoppers mainly damage high-quality forage grasses such as *Leymus chinensis*, *Stipa grandis*, and *Elymus sibiricus*. Due to the climate and environmental conditions of the grassland in Inner Mongolia, grasshoppers can usually cause a disaster within one generation annually and spend the winter in the soil with eggs. Larvae emerge from the soil in April–May, and swarming occurs in June–July. The normal length of survival is generally about 90 days [36]. Among the factors that affect the reproduction of grasshoppers in

Inner Mongolia, temperatures, soil types, and moisture are critical. For example, when the temperature in spring is about 12–18 °C, eggs hatch, and in summer, temperatures of 25–40 °C are conducive to the rapid maturation of grasshoppers. Neutral sandy soil or loam is the ideal egg-laying environment for grasshoppers, and a soil moisture of 15–20% is necessary for egg hatching and the growth of weak insects [37].

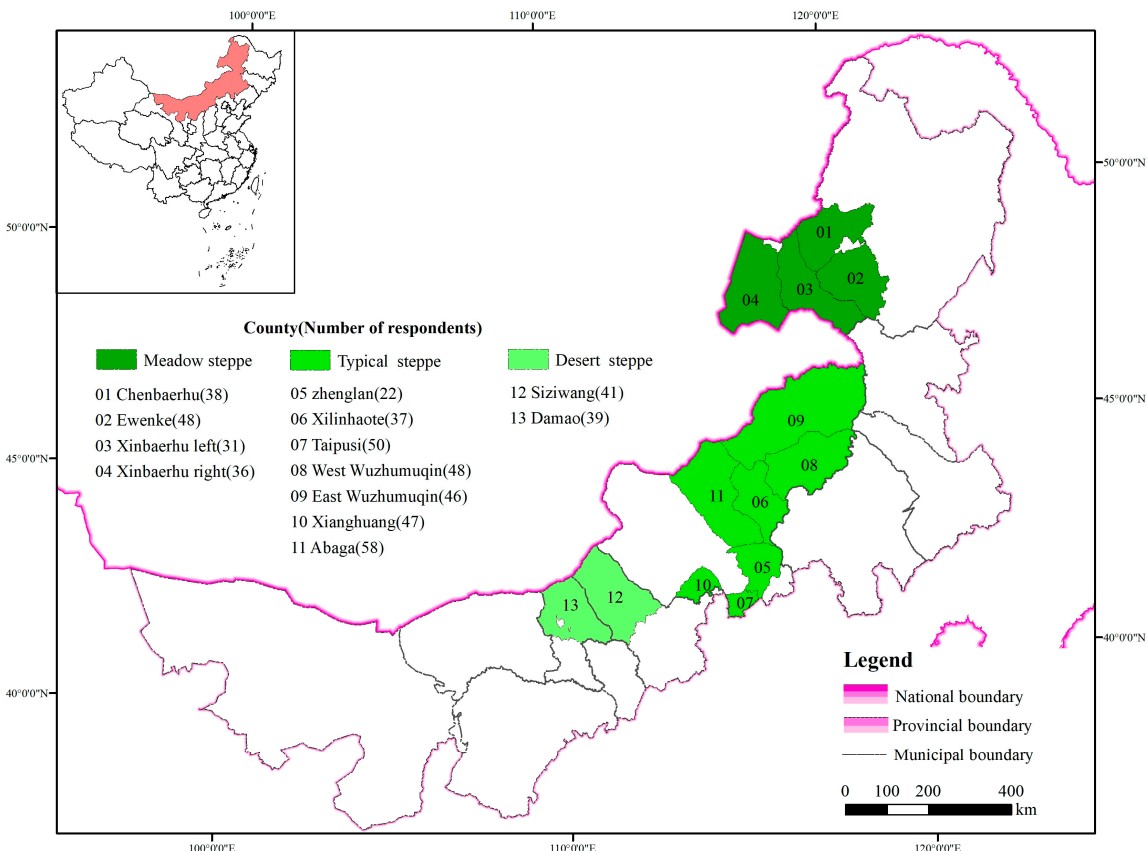

**Figure 2.** Map of the study area.

## 2.2. Data Collection

The data was obtained via a face-to-face household survey using structured questionnaires in 2020. Random and purposive sampling were jointly employed in the survey. Before sampling, we purposely chose 13 counties that had experienced locust infestations in the recent 5 years from 2016 to 2020 based on the government's monitoring reports. Then, at least three towns in each target county were selected according to the frequency of locust disasters, from high to low. Next, more than two villages were randomly selected in each targeted town. All the households in each village were categorized into three groups based on their household incomes. Two or three households were randomly chosen in each group to make sure they represented local households with different financial statuses. The herders who participated in the survey were able to identify locusts fairly accurately. In addition, we invited local government staff involved in locust control to conduct interviews with herdsmen, who could help and guide herdsmen to complete the questionnaire. The interview among the respondents lasted 45–60 min. Cumulatively, 550 households from 13 counties were surveyed, among which 541 valid samples were used for analysis, including 153 from meadow steppe, 308 from typical steppe, and 80 from desert steppe.

The questionnaire included three parts: (1) Basic information about the respondents and their families, including gender, age, ethnicity, education levels, and family size. (2) Livestock production information, including pasture areas and utilization, numbers and structures of livestock, possession of fixed assets, knowledge of grassland, and pasture protection behavior. (3) The frequency and losses from grasshopper disasters in their own

grasslands and surrounding areas; the impact of grasshopper disasters on their production and lives.

The basic characteristics of the respondents are shown in Table 1. To collect more accurate information, the heads of households, usually males and under 60, were chosen as target respondents purposefully. In terms of education levels, junior high school and elementary school levels were most common, accounting for 38.45% and 28.65%, respectively. The average area of grasslands used by a household was 366 ha, and the average number of livestock was 418 se. The grassland area and the number of livestock in the meadow steppe were larger than those in the typical steppe and desert steppe.

**Table 1.** Basic information of the interviewees.

| | Category | Total | Meadow Steppe | Typical Steppe | Desert Steppe |
|---|---|---|---|---|---|
| Nation (%) | Mongolian | 84.1 | 83.66 | 86.36 | 76.25 |
| | Han | 13.68 | 9.15 | 13.64 | 22.50 |
| | Other | 2.22 | 7.19 | 0.00 | 1.25 |
| Age (%) | 16–45 | 45.29 | 56.21 | 43.83 | 30.00 |
| | 45–60 | 43.81 | 31.37 | 47.40 | 53.75 |
| | >60 | 10.91 | 12.42 | 8.77 | 16.25 |
| Gender (%) | Male | 94.82 | 98.69 | 93.18 | 93.75 |
| | Female | 5.18 | 1.31 | 6.82 | 6.25 |
| Education level (%) | Illiteracy | 9.8 | 7.84 | 1.62 | 45.00 |
| | Primary school | 28.65 | 32.68 | 27.27 | 26.25 |
| | Middle school | 38.45 | 50.33 | 35.06 | 28.75 |
| | High school/vocational technical high school | 16.64 | 3.92 | 27.27 | 0.00 |
| | Undergraduate degree and above | 6.47 | 5.23 | 8.77 | 0.00 |
| Operating grassland area (ha) | | 372 | 491 | 331 | 314 |
| Livestock number (se) | | 418 | 500 | 420 | 261 |

Note: Sheep equivalent (se) is an 45 kg adult sheep which consumes 1.8 kg hay per day. In the conversion of different animals in Inner Mongolia, 1 sheep equals 1 se, 1 goat equals 0.8 se, 1 cow or beef cattle equals 5 se, 1 horse equals 6 se, and 1 camel equals 7 se.

### 2.3. Statistical Models

The data was analyzed using Stata 15.1 software (Stata Corp, StataCorp LLC, Raleigh, NC, USA). Two dependent variables, namely "the frequency of grasshopper disasters" and "the losses of grasshopper disasters", were selected to represent grasshopper disasters. The generalized Poisson model and the OLS regression model were used, respectively, to estimate the influences.

The generalized Poisson regression model is a generalization of the standard Poisson regression model and is widely adopted in various fields, including agriculture and health care due to its advantages in dealing with over- and under-dispersed data [38,39].

Suppose $Y_i$ is a count response variable, and follows a generalized Poisson distribution, the probability mass function (PMF) of $Y_i$, $i = 1, 2, \ldots, n$ according to Wang and Famoye [40] is specified as:

$$f(y_i) = P_r(Y_i = y_i) = \left(\frac{\lambda_i}{1 + \alpha\lambda_i}\right)^{y_i} \frac{(1 + \alpha y_i)^{y_i-1}}{y_i!} exp\left(-\frac{\lambda_i(1 + \alpha y_i)}{1 + \alpha\lambda_i}\right), \ i = 0, 1, 2, \ldots, n \quad (1)$$

The mean and variance of $Y_i$ are mathematically specified as:

$$E(Y_i|x_i) = \lambda_i, Var(Y_i|x_i) = \lambda_i(1 + \alpha\lambda_i)^2 \quad (2)$$

In Equations (1) and (2), $x_i\left(1, x_{i1}, \cdots, x_{i(k-1)}\right)^{\mathrm{T}}$ is the independent variable, $\lambda_i = \lambda_i(x_i) = exp(x_i^T\beta)$ is the perturbation term, and $\beta = (\beta_1, \beta_2, \cdots, \beta_k)^{\mathrm{T}}$ is the regression coefficient. $\alpha$ is the dispersion parameter. The maximum likelihood method is used to calculate the estimates of $\beta$ in the generalized Poisson regression model.

### 2.4. Variable Selection and Data Description

The frequency and losses of grasshopper disasters perceived by herders in the past 15 years (2005–2020) were chosen as explained variables, while the explanatory variables (Table 2) were selected based on relationships with livestock production and grassland grasshopper disasters. Data on operating grassland areas, renting grassland areas, the number and structure of livestock, feeding forage, and grazing rates were harvested to represent herders' animal husbandry production behaviors [41]. In addition, considering the influence of guidelines and grassland quality, the effect of guidelines and herders' knowledge of grasslands were selected as explanatory variables.

**Table 2.** Locust occurrence and influencing factors.

| Variable | Definition | Mean | Standard Deviation | Minimum | Maximum |
|---|---|---|---|---|---|
| | Explained variable | | | | |
| The frequency of grasshopper disasters ($y_1$) | 1 = no occurrence, 2 = less than 3 times, 3 = 3–6 times, 4 = 6–10 times, 5 = every year | 2.58 | 1.22 | 1 | 5 |
| The losses of grasshopper disasters ($y_2$) | Percentage of grassland area perceived by herders to have been damaged by grasshopper disasters to operating grassland area in the last 15 years. | 18.23 | 14.41 | 0 | 60 |
| | Explanatory variable | | | | |
| Operating grassland area ($x1$) | Operating grassland area refers to the area used by herdsmen, including grassland contracted from local community and renting grassland (ha). | 372 | 389 | 17 | 3066 |
| Renting grassland area ($x2$) | Renting grassland area refers to the area rented from other grassland (ha). | 91 | 259 | 0 | 2600 |
| Livestock number ($x3$) | The number of livestock raised in summer. Livestock number is converted into se. | 418 | 358 | 15 | 1821 |
| Livestock structure ($x4$) | 1 = Pure small livestock (sheep, goat) or pure large livestock (beef cattle, cow, horse), 2 = Mixed large livestock and small livestock | 1.69 | 0.46 | 1 | 2 |
| Feeding forage ($x5$) | Amount of forage supplemented per sheep equivalent per year (kg/se). | 229 | 193 | 0 | 756 |
| Grazing rate ($x6$) | Ratio of the number of livestock actually grazed to the number of livestock in a sustainable grazing scheme per unit area of pasture. | 1.46 | 1.54 | 0.02 | 12.45 |
| Policy intensity ($x7$) | 1 = grassland-livestock balance, 2 = grassland-livestock balance + rest grazing, 3 = grassland-livestock balance + rest grazing + closure against grazing or grassland-livestock balance + closure against grazing | 2.18 | 0.73 | 0 | 3 |
| Grassland perception ($x8$) | 1 = very bad, 2 = relatively poor, 3 = general, 4 = fine, 5 = very good | 2.79 | 1.04 | 1 | 5 |

The grazing rate was used to characterize the intensity of grazing, which is the ratio of the number of livestock actually grazed per hectare to that in a sustainable grazing scheme [42]. Thanks to the recommended forage-livestock balance, well-perceived or sustainable numbers of livestock could be easily accessed in research regions. The grazing rate ranged from 0.02 to 12.45 and was considered proper within 0.8 to 1.2 (see Appendix A for data).

Policy intensity refers to policies about subsidies and rewards implemented by the Chinese government for grassland ecological protection in grasslands, which aim to protect the grassland ecological environment through a series of measures such as the ban on

grazing, the balance between forage and livestock, and the cessation of grazing. The effect of guidelines in this study refers to the combination of these three measures in the ranches. It is believed that more measures mean better protection.

The reliability and validity of the survey data were tested by SPSS26.0. Cronbach's alpha values for all dimensions were above 0.812, indicating acceptable reliability. The KMO value was 0.733 > 0.7, and the Bartlett test significance was 0.000, which indicated the reliability and validity of the data [43].

## 3. Results

### 3.1. Frequency of Grasshopper Disasters Perceived by Herders

Grasshopper disasters were considered severe when they occurred more than three times. Forty-two percent of the respondents, 225 households, reported that grasshopper disasters had occurred more than three times in the past 15 years, and the average frequency of grasshopper disasters was 3.97 (Figure 3 and Appendix A). These high-frequency observations were predominantly located in Damao Banner, Xianghuang Banner, and West Wuzhumuqin Banner. A total of 316 (58%) herders reported less than three grasshopper disasters in the past 15 years, including 227 "less than 3 times" and 89 "no occurrence". In the four counties in the meadow steppe, especially the Xinbaerhu Left Banner and the Xinbaerhu Right Banner, the frequency of grasshopper plagues was less than three times. A comparison between different grassland types showed that herders from desert steppe and typical steppe experienced more frequent grasshopper disasters than those from meadow steppe.

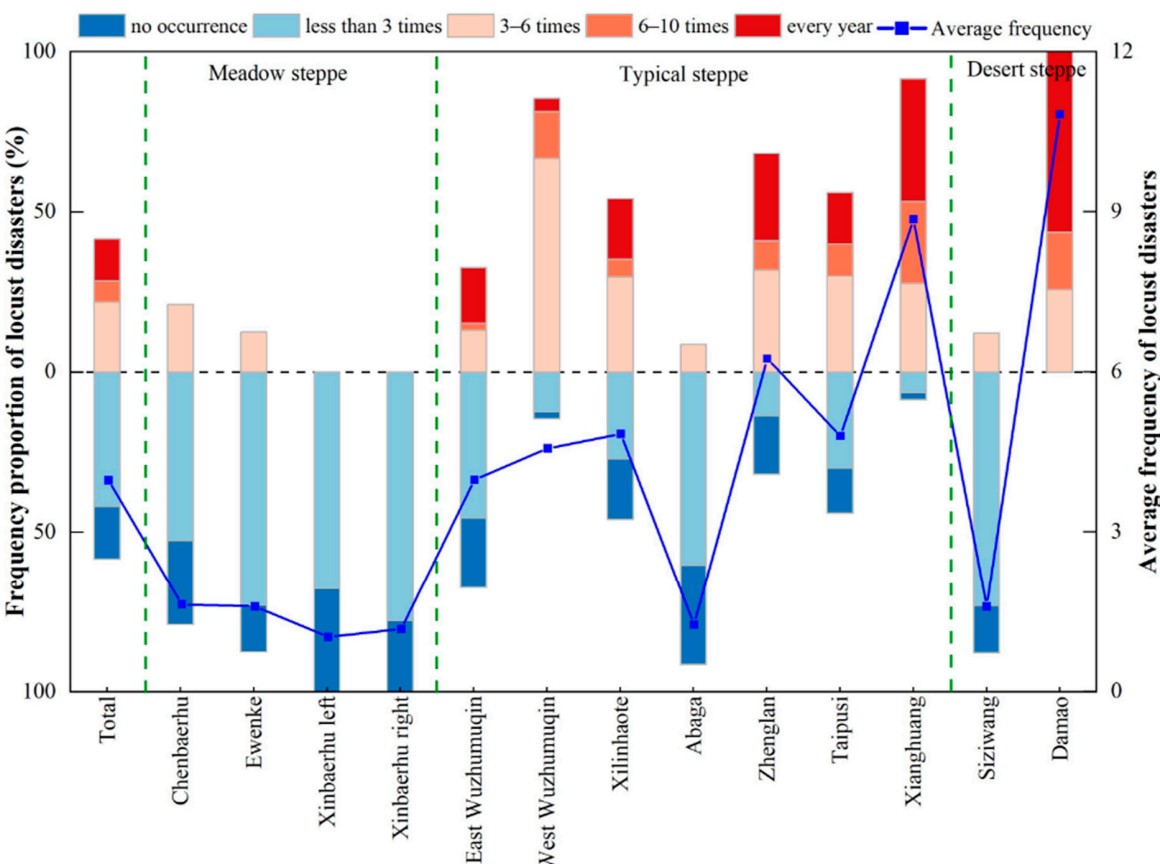

**Figure 3.** Grasshopper disaster frequency during 2005 to 2020 perceived by herders. Note: The cumulative columns in the figure represent the proportion of grasshopper disasters frequency, and the sum is 100%. Below the horizontal line are "less than 3 times" and "no occurrence", above the horizontal line are "3–6 times", "6–10 times", and "every year". The broken line is the average of grasshopper plague frequency in each county. The calculation process is in Appendix A.

There were significant differences in the grasshopper disaster frequency perceived by herders under different grazing rates (Figure 4). With the increase in grazing intensity, the frequency of grasshopper disasters decreased and then increased. As is shown in Figure 4a, the average frequency of grasshopper disasters perceived by herders was 2.32 when the grazing rate was 0.4–0.8, and the proportion occurring more than three times was 16.67%. The highest frequency was found when the grazing rate was ≥1.6, the average frequency was 6.44, and the proportion of cases of more than three times was 67.88%. The frequency of grasshopper disasters in meadow steppe was the lowest, and with the increase in the grazing rate, the frequency trended up. The frequency in a typical steppe was relatively high, decreasing first and then increasing. The desert steppe had the highest frequency of grasshopper disasters, and there was obviously no fixed pattern concerning frequency under different grazing rates.

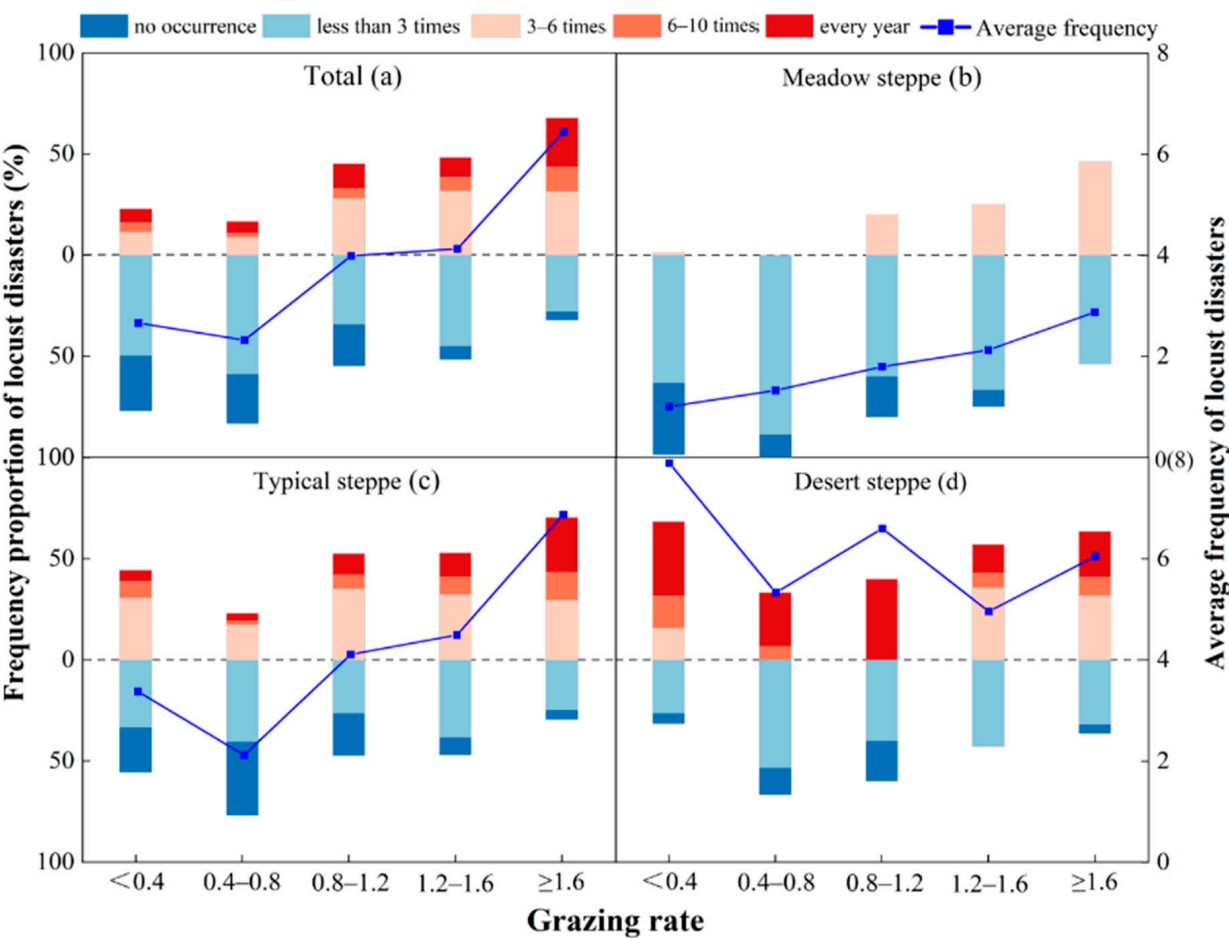

**Figure 4.** Map of grasshopper disaster frequency under different grazing rate. Note: (**a–d**) shows the total sample, meadow steppe, typical steppe and desert steppe, respectively. The columns and lines accumulated in the figure are the proportion and average of grasshopper plague frequency, respectively.

### 3.2. Losses from Grasshopper Disasters Perceived by Herders

The average grassland area damaged by grasshopper disasters in the study area was 49.00 hectares per household, accounting for 18.23% of the operating grassland area in the past 15 years (Figure 5). A total of 267 (49.35%) households suffered more serious losses than the average (49.00 ha), which indicated a symmetric distribution of grassland losses among all surveyed households. The most serious damage was in Damao Banner, where the average area and proportion of damage per household were 102 ha and 37%, significantly higher than in other areas. The average area of meadow steppe, typical steppe,

and desert steppe damaged per household was 42.93 ha, 49.90 ha, and 73.24 ha, respectively, and the proportion of damaged grassland was 7.76%, 21.28%, and 25.31%, respectively.

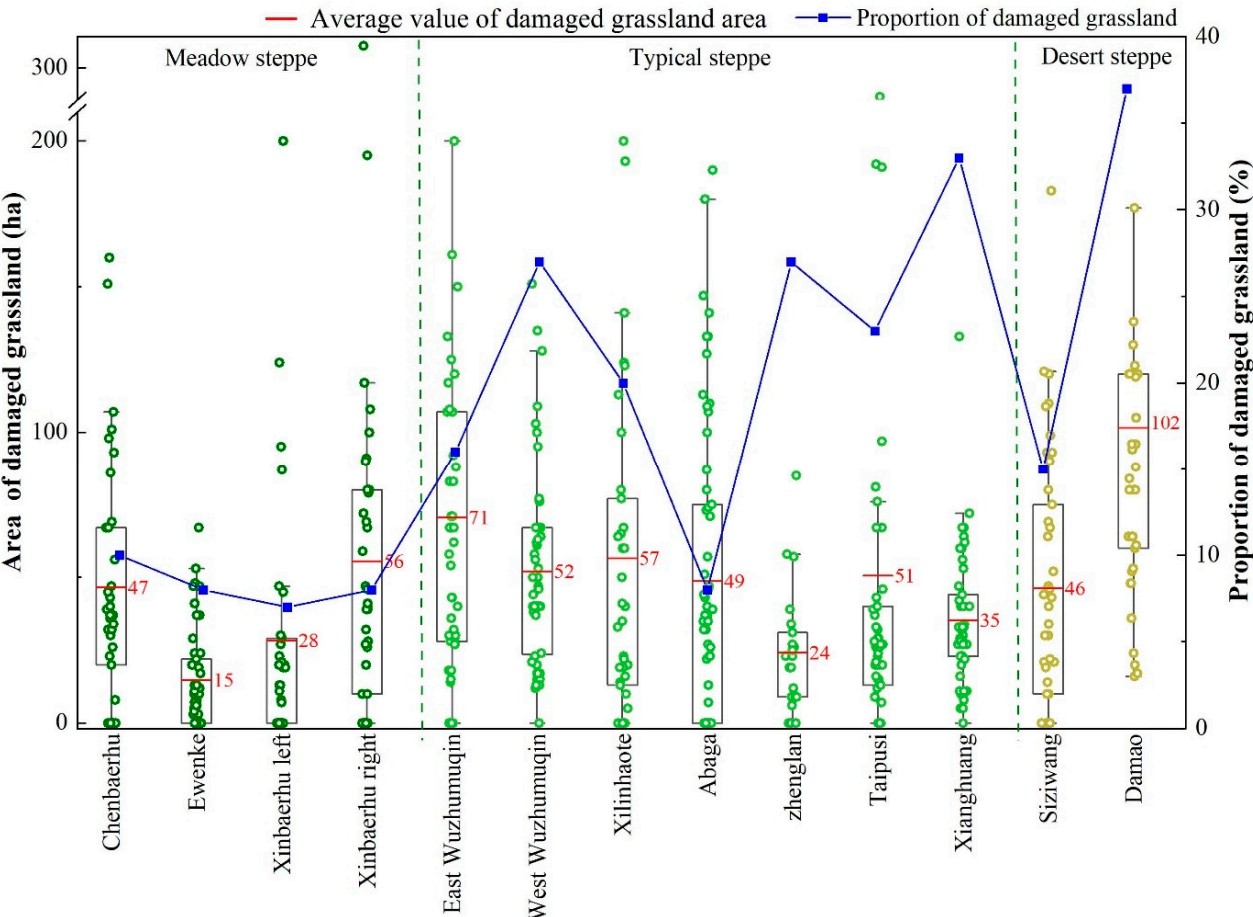

**Figure 5.** Map of the losses affected by grasshopper disaster. Note: The box plot in the figure represents the area of grassland damaged by grasshopper disaster, and the broken line represents the average proportion of grassland damaged by grasshopper disaster.

The results also showed that the grassland area damaged by grasshopper disasters increased with the grazing rate (Figure 6). It can be seen from Figure 6a that when the grazing rate was <0.4 and 0.4–0.8, the area and proportion of grassland damaged were the lowest. When the grazing rate was 1.2–1.6, the highest grassland area was 66 ha. When the grazing rate was >1.6, the proportion of grasslands damaged was the highest (27.61%). Among the three types of changes in grasslands, the change in the grassland area damaged did not follow any known pattern, but the change in the proportion of the grassland areas damaged conformed with the change in the total study sample.

### 3.3. The Impact of Production Behavior on Grasshopper Disasters

The regression results showed that operating grassland area, renting grassland area, feeding forage, grazing rate, and the effect of guidelines impacted the frequency of grasshopper disasters, as was expected (Table 3). In the generalized Poisson model, the Pseudo $R^2$ is 0.2439, with a LR chi2 of 1129.84, a Prob > chi2 of 0.0000, and a Log likelihood of −617.4663. In the OLS regression model, the $R^2$ is 0.3972, with Prob > F at 0.0000, and Root MSE is 11.297. The above results all indicated a strong model fit to the data.

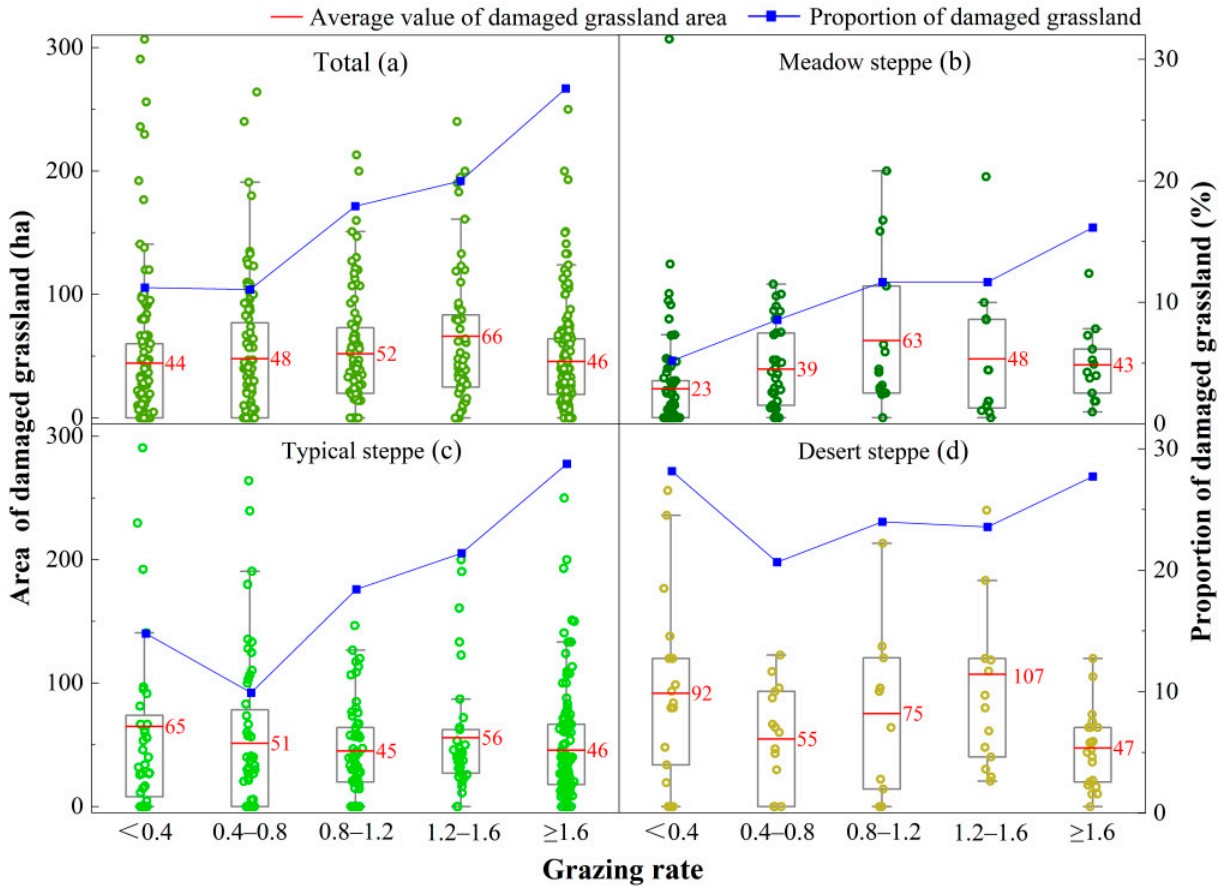

**Figure 6.** Map of grasshopper disaster losses under different grazing rate. Note: (**a**–**d**) shows the total sample, meadow steppe, typical steppe and desert steppe, respectively. The box plot in the figure represents the area of grassland damaged by grasshopper disaster, and the broken line represents the average proportion of grassland damaged by grasshopper disaster.

**Table 3.** The influences of production behavior on grasshopper disasters.

| Explanatory Variable | The Frequency of Grasshopper Disasters ($y_1$) | | The Losses of Grasshopper Disasters ($y_2$) | |
|---|---|---|---|---|
| | Coefficient | Standard Error | Coefficient | Standard Error |
| Operating grassland area ($x1$) | −0.0002 *** | 0.0001 | −0.0066 *** | 0.0025 |
| Renting grassland area ($x2$) | 0.0003 *** | 0.0000 | 0.0055 ** | 0.0029 |
| Livestock number ($x3$) | 0.0001 * | 0.0001 | 0.0023 | 0.0019 |
| Livestock structure ($x4$) | −0.0159 | 0.0355 | −1.4705 | 1.2530 |
| Feeding forage ($x5$) | −0.0002 ** | 0.0001 | −0.0063 * | 0.0033 |
| Grazing rate ($x6$) | 0.0567 *** | 0.0132 | 3.5833 *** | 0.5327 |
| Policy intensity ($x7$) | 0.0413 * | 0.0212 | 3.8467 ** | 0.6588 |
| Grassland perception ($x8$) | 0.0157 | 0.0160 | 0.8371 | 0.5169 |
| Typical steppe | 0.3632 *** | 0.0459 | 7.0364 *** | 1.1902 |
| Desert steppe | 0.4513 *** | 0.0579 | 11.3104 *** | 2.0198 |
| _cons | 0.8537 | 0.1034 | 1.2401 | 3.4793 |
| | Log likelihood = −617.4663<br>Prob > chi2 = 0.0000<br>Pseudo R2 = 0.2439<br>LR chi2(20) = 1129.84 | | Prob > F = 0.0000<br>R-squared = 0.3972<br>Root MSE = 11.297 | |

Note: Other variables omitted for lack of statistical significance were herder ethnicity, gender and household size. * means significant at the 10 percent level. ** means significant at the 5 percent level. *** means significant at the 1 percent level.

Operating grassland areas had a significant negative impact on grasshopper disasters, indicating that the larger the grassland area operated by herders, the lower the frequency of grasshopper disasters. The larger the area of operating grasslands, the lower the intensity of grassland use. In addition, livestock production activities will be more diverse, and herders will be in a position to rotate grazing and mow hay, which will benefit the protection of grasslands.

Renting grassland areas affected grasshopper disasters positively, indicating that the more renting grassland areas herders had, the higher the frequency of grasshopper disasters in their ranch. Grassland renting was a popular activity in research regions and was therefore incorporated into the model. Grasslands were contracted to households for 30 years so that herders could expand production scales by renting grasslands and purchasing forages. Therefore, renting has become an important means of livestock production. The results showed that although herders reduced grazing rates by renting grasslands, the intensity of use of rented grasslands was higher than that of contracted grasslands, which led to the occurrence of grasshopper disasters.

The quantity and structure of livestock influence grassland grazing stress directly. In this study, these two indicators proved to have positive and negative effects on grasshopper occurrence, respectively. Livestock numbers were correlated with the frequency of grasshopper disasters at the 0.1 level, but not with losses from grasshopper disasters. The livestock structure was not statistically significant.

Feeding forage had significant negative effects on the frequency and loss of grasshopper disasters at 0.05 and 0.1 levels, respectively, indicating that the more forage herders supplemented the livestock, the lower the frequency of grasshopper disasters on their ranch. In this research, feeding forages were defined as forages supplemented per sheep equivalent per year. To be more specific, they referred to forages that were purchased from markets, which were resources from outside the rangeland that herders used in their livestock production. It was believed that the less grassland herders used in their production, the fewer grasshoppers could be observed.

The grazing rate had a significantly positive effect on grasshopper disasters, which means the more prevalent overgrazing, the more frequent grasshopper disasters became. In this study, the grazing rate refers to the ratio of the actual number of livestock per unit area of grassland in a year to the acceptable number of livestock. When the grazing rate is higher, the intensity of grassland utilization is greater, which will have a negative impact on the ecological stability and health of grassland, so a grasshopper plague is more likely.

Policy intensity had positive effects on the frequency and losses of grasshopper disasters at 0.1 and 0.01 significant levels, respectively. The results showed that grasshopper plagues became more frequent and devastating with the increase in related guidelines. One possible reason was that places under more mandates, especially grasslands where overgrazing was banned, were mostly heavily degraded areas and therefore more prone to grasshopper disasters.

In addition, based on the statistics of Pearson correlation coefficients between grasshopper disasters and indicators of production behavior (Figure 7), it could be found that each production behavior had an impact on grasshopper disasters as a result of interactions. Operating grassland areas, renting grassland areas, livestock numbers, and structures were significantly positively correlated. The grazing rate was significantly negatively correlated with the operating grassland area, renting grassland area, and feeding forage, but significantly positively correlated with livestock numbers. The amount of knowledge of grasslands was significantly negatively correlated with livestock structures and feeding forages.

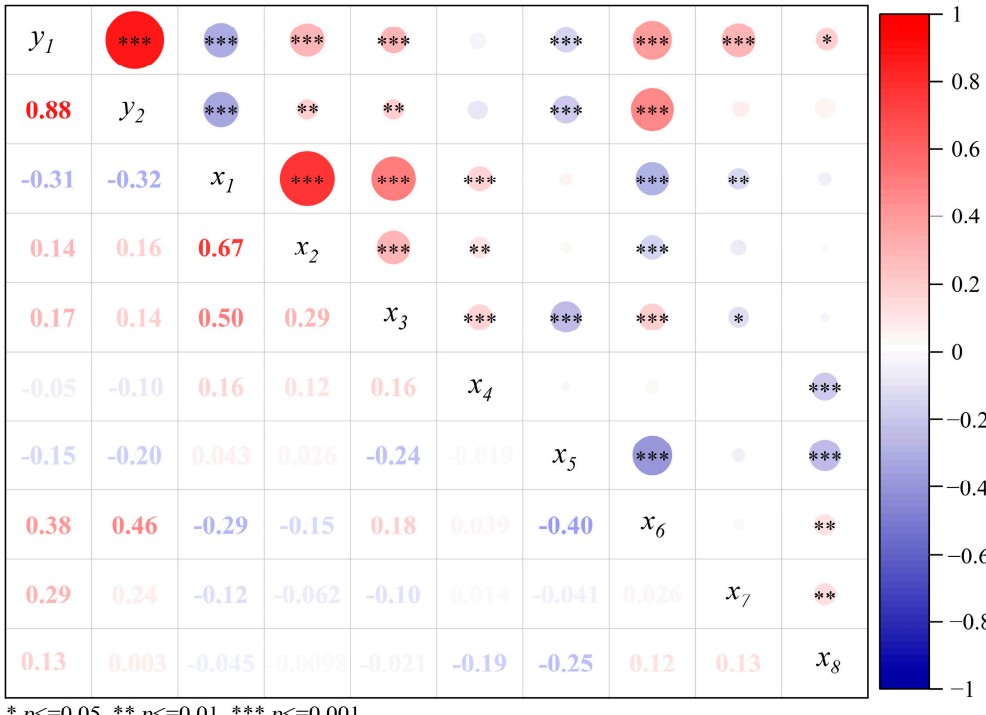

* $p<=0.05$  ** $p<=0.01$  *** $p<=0.001$

**Figure 7.** Correlation between grasshopper disasters and indicators of production behavior.

## 4. Discussion

### 4.1. How Grazing Influences Grasshopper Disasters

This social survey confirmed that overgrazing induces grasshopper disasters. The biophysical and ecological factors are of two types. Firstly, overgrazing affects locust species composition by affecting community changes in vegetation. Secondly, overgrazing increases dry, bare, land, which favors the laying of eggs by grasshoppers [10]. Finally, overgrazing with massive livestock degrades grasslands towards low-nitrification plants, which will provide favorite diets and consequently lead to the outbreak of Asian grasshoppers [14].

On the contrary, a proper grazing regime would reduce the outbreak of grasshoppers by preserving the vegetation canopy and reducing the bare land at the critical period of grasshopper reproduction [44]. Although not statistically significant, diverse livestock structures had negative influences on grasshopper disasters, which means that, compared with single species, raising diversified livestock would lower the risk of grasshopper disasters.

On the other hand, people should notice that a proper grazing regime is not equal to a low stocking rate. From Figures 4 and 6, we can see that the lowest points of both occurrence frequency and losses from grasshopper disasters appeared at a moderate grazing rate rather than the lowest grazing rate. A large number of previous studies have also shown that moderate grazing can protect grassland from degradation, contribute to the diversity of grasshopper communities, and have a low proportion of pest species [16,45]. This can be explained by the moderate disturbance theory, which suggests that a moderate disturbance on grassland can maintain a high species diversity by maintaining the diversity of grasshopper disasters, reducing their abundance, and avoiding the outbreak of grasshopper disasters.

Furthermore, the "degree" of moderate disturbance is different in different grassland types. According to previous studies, grasslands with higher biomass and diversity and better health conditions are more tolerant to grazing utilization [46–50]. Munkhzul et al. [51] found that grazing had a strong impact on the vegetation of dry steppe, desert steppe, and alpine steppe but a weak impact on meadow steppe and alpine steppe with mild steppe

climate and high productivity, which indirectly explained why grasshoppers in meadow steppe were less frequent and detrimental than in other steppes [44,52].

### 4.2. Why Operating Grassland Area Matter

This research not only reconfirmed that overgrazing impels grasshopper disasters but also empirically proved that herders with larger operating grassland suffered less from grasshopper disasters. By enlarging the operating grassland area, herders can employ relatively reasonable grazing regimes like rotational grazing and resting part of the grasslands in critical seasons, which contribute to improving the grassland ecosystem and reducing grasshopper disasters.

Furthermore, this becomes more relevant when dissecting the "operating grassland". According to the survey, operating grassland mainly included two parts: contracted grasslands and rented grasslands. After the grasslands were contracted to herders' households in the 1980s and 1990s, the contracted area was stabilized. Being restricted by the flexibility of accessible grasslands compared with nomadism, herders turned to the grassland renting market to meet their demand for expanding production scale. In total, 29.30% of households in this survey had renting experiences during the last 5 years, and the average rental area was 310.93 ha, accounting for 50.49% of their operating grassland area. In this study, herders who rented grassland had a larger production scale and a lower grazing rate, but the model results showed a significant positive correlation between renting grassland and grasshopper disasters.

Grassland renting is not a new topic; however, its influences on grasshopper disasters have seldom been discussed. Jimoh et al. [53] reported that renting in more grasslands decreased herders' stocking rate; however, in his previous study, he reported that grassland rent-in is one of the principal factors that induce overgrazing [54]. As these studies were carried out at the household level, rented grassland and non-rented grassland were not treated separately. Lu et al. [55] proved that, compared to the rented grasslands, those that have not been rented had better grassland greenness. It can then be inferred that herders expand their operating grasslands by renting in grasslands, reducing their stocking rate at the household level. However, the rented grasslands are not used equally as the owned grasslands; they are more likely to be overgrazed. In other words, by renting grasslands, herders' own grasslands are preserved, which consequently contributes to better vegetation and lower grasshopper frequency.

### 4.3. The Influences of Supplementation to Grasshopper Disasters

Supplementation of range livestock is more and more popular all over the world [56]. Though originally adopted as an adaptive strategy for disasters, it is now widely used to improve grassland livestock production efficiency. In the study region, the supplementary period is expanded to 4–7 months per year. But its ecosystem influences have not been reported yet. In this study, supplementation was represented by forage supplemented per sheep equivalent per year. It is easy to understand that the more supplementation, the less grassland is used, and as a result, grassland is preserved and grasshopper disasters are reduced.

Although supplementation of livestock in rangeland decreased the occurrence and damage of grasshopper disasters in this research, its influences on ecosystems and even grasshopper disasters in the long run have to be cautious. The supplementary forages from farming and the farming-pastoral region decoupled the local rangeland social-ecological system [57] to a certain extent, which might give rise to new ecological risks people had not expected.

*4.4. Limitations and Future Work*

This paper has quantitatively analyzed the relationships between production behaviors of livestock husbandry and grasshopper disasters at a micro-herders' level, which is an important supplement to the case study on the relationship between grasshopper disasters and humans and also an important verification of previous studies. However, there are still some limitations. This research was conducted based on the occurrence of grasshopper disasters rather than a controlled experiment designed to simulate the occurrence of grasshopper disasters. Consequently, the findings were based on herders' observations and experiences rather than on real causal-effect research. Therefore, the reliability of the results might be challenged by some authors. But the aim of this research was to shed light on the relationship between herders' grazing activities and grasshopper occurrences. In this connection, herders' extensive observation is more reliable than control experiments, given the complexity of rangeland management. To improve the accuracy of research, more variables related to grazing activity, soil, and vegetation could be adopted in the future.

**5. Conclusions**

This study revealed the relationships between herders' livestock production behaviors and the outbreak of grasshopper disasters and further enriched the empirical cases of human activities and grasshopper disasters from the perspective of micro-herders. This study suggested that herders' livestock production activities can influence grasshopper outbreaks directly or indirectly by adjusting stocking intensity on grasslands. Renting in grasslands and supplementing livestock can lower locust risks by reducing stocking intensity, while overgrazing is likely to induce grasshopper occurrences by increasing stocking intensity.

Herders are grassland operators, so they are critical to the prevention and control of grassland grasshopper disasters. This study believes that herders should optimize the production mode of animal husbandry. They should reduce the pressure on natural grasslands through livestock reduction and supplemental feeding and focus on improving the quality of livestock in order to achieve win-win in terms of both livestock reduction and income generation, as well as ecological and economic balance.

**Author Contributions:** S.L.: Investigation, Data curation, Conceptualization, Writing—Original Draft Preparation, Writing—Review and Editing. M.C.: Methodology, Conceptualization, Writing—Original Draft, Writing—Review and Editing. P.L.: Investigation, Conceptualization, Writing—Review and Editing, Funding acquisition. T.B.: Investigation, Data curation. X.H.: Investigation. G.Y.: Data curation. M.C. and S.L. are the first authors of this paper because of the same contribution to this work. All authors have read and agreed to the published version of the manuscript.

**Funding:** This work was supported by [the Science and Technology Plan Projects of Inner Mongolia Autonomous Region] grant number [2022YFDZ0040], [Innovation Fund of Inner Mongolia Academy of Agricultural and Animal Husbandry Science] grant number [2022QNJJM01], and [the China Forage and Grass Research System] grant number [CARS-34].

**Data Availability Statement:** The data used in this study are questionnaire survey data. Due to the need for further in-depth mining and analysis of the data presented in this paper, the data cannot currently be shared publicly. Researchers interested in learning more about the data details are welcome to contact the corresponding author to discuss potential opportunities for data access.

**Acknowledgments:** We are also grateful to the local officials of the 13 counties in the study area who helped us with our field work, as well as the 541 herders who provided us with valuable data.

**Conflicts of Interest:** We declare that we do not have any commercial or associative interest that represents conflicts of interests in connection with the work submitted, and manuscript is approved by all authors for publication. I would like to declare on behalf of my co-authors that the work described was original research that has not been published previously, and not under consideration for publication elsewhere, in whole or in part.

**Appendix A**

*Appendix A.1 Calculation of the Average Frequency of Grasshoppers*

The frequency of grasshopper disasters perceived by herders in the past 15 years (2005–2020) in this study (Table A1). In order to better analyze the occurrence of grasshopper disasters in each county, the average frequency of grasshopper disasters in each county was calculated according to Equation (A1).

$$\text{Average frequency} = (\text{no occurrence} \times 0 + \text{less than 3 times} \times 1.5 + \text{3-6 times} \times 4 + \text{6–10 times} \times 7.5 + \text{every year} \times 15)/\text{Number of respondents} \tag{A1}$$

**Table A1.** The primary grasshopper species damaging grasslands in the Inner Mongolia grassland area.

| Species | Damaged Grassland Types | | | Size Category | Occurrence Period | Control Standards (Density of Insects per Square Meter) |
|---|---|---|---|---|---|---|
| | Meadow Steppe | Typical Steppe | Desert Steppe | | | |
| *Oedaleus decorus* (Germar) | + | + | + | Medium | mid-season | 15 |
| *Dasyhippus barbipes* (Fischer-Waldheim) | + | + | + | Small | early season | 25 |
| *Myrmeleotettix palpalis* (Zubowsky) | + | + | + | Small | early season | 25 |
| *Angaracris barabensis* (Pallas) | + | + | + | Medium | mid-season | 15 |
| *Bryodema luctuosum* (Stoll) | - | + | + | Medium | early season | 15 |
| *Bryodemella tuberculatum dilutum* (Stoll) | + | + | - | Medium | late-season | 15 |
| *Pararcyptera microptera meridionalis* (Ikonnikov) | - | + | - | Medium | mid-season | 15 |
| *Chorthippus dubius* (Zubovsky) | - | + | - | Small | late-season | 25 |
| *Calliptamus abbreviatus* Ikonnikov | + | - | + | Medium | late-season | 15 |

Note: The "+" indicates that this species of grasshopper is the dominant species in the corresponding grassland area. Size category definition: According to the "Main Pest Control Standards" issued by the China National Forestry and Grassland Administration, small grasshoppers, medium grasshoppers, and large grasshoppers are identified as species with an adult body length less than 20 mm, 20–40 mm, and more than 40 mm, respectively. According to survey data, the early season species are active in April–May; mid-season species are active from May to July; late-season species are active from July to September. Control standards: According to the "Main Pest Control Standards" issued by the China National Forestry and Grassland Administration [58].

*Appendix A.2 Analysis of Grazing Rate*

As a concept to measure the balance of grass and livestock, the grazing rate is receiving more and more attention. It refers to the ratio of the number of livestock actually grazed to the number of livestock in a sustainable grazing scheme per unit area of pasture. The grazing rate ranged from 0.02 to 12.45, with a mean value of 1.46, which is the state of overload (Figure A1). The typical steppe is the most severely overloaded area. The overload rate of all the counties is higher than that of other grassland types. The grazing rate of Zhenglan Banner and West Wuzhumuqin Banner is the largest in the survey area, with an average of 2.81 and 2.30, respectively. The grazing rate of meadow grassland is the lowest, and the mean value of the four counties is less than 1, showing no overloading.

In order to observe the difference in grasshopper disaster occurrence under different grazing rates, which were divided into five levels. The grades ranged from <0.4, 0.4 to 0.8, 0.8 to 1.2, 1.2 to 1.6, and ≥1.6, accounting for 24.21%, 19.04%, 15.16%, 11.09%, and 30.50% of the total samples, respectively (Table A2). The pasture utilization intensity is reasonable in the range of 0.8–1.2 grazing rate.

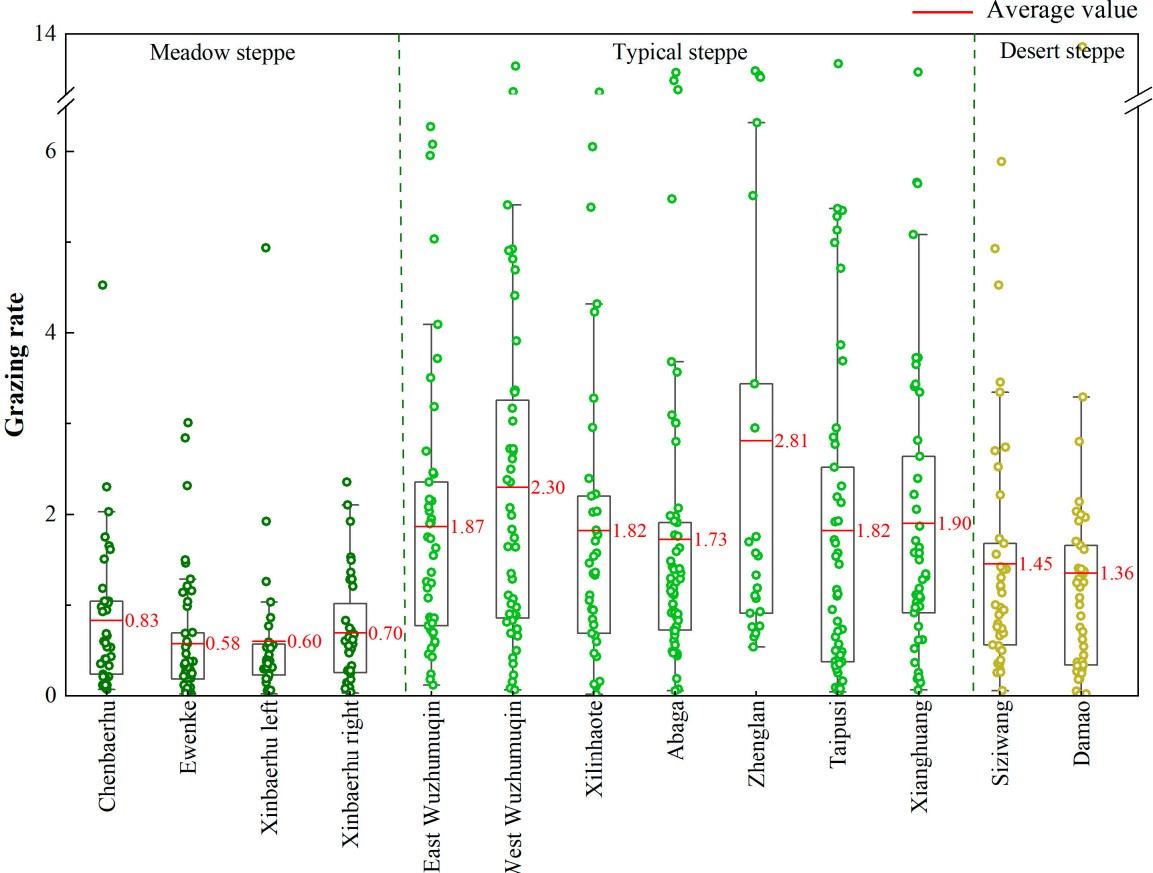

**Figure A1.** Map of the grazing rate.

**Table A2.** Grasshopper disaster frequency from 2005 to 2020 perceived by herders.

| County | No Occurrence | Less than 3 Times | 3–6 Times | 6–10 Times | Every Year | Average Frequency |
|---|---|---|---|---|---|---|
| Total | 89 | 227 | 118 | 36 | 71 | 3.97 |
| Chenbaerhu | 10 | 20 | 8 | 0 | 0 | 1.63 |
| Ewenke | 7 | 35 | 6 | 0 | 0 | 1.59 |
| Xinbaerhu left | 10 | 21 | 0 | 0 | 0 | 1.02 |
| Xinbaerhu right | 8 | 28 | 0 | 0 | 0 | 1.17 |
| East Wuzhumuqin | 10 | 21 | 6 | 1 | 8 | 3.98 |
| West Wuzhumuqin | 1 | 6 | 32 | 7 | 2 | 4.57 |
| Xilinhaote | 7 | 10 | 11 | 2 | 7 | 4.84 |
| Abaga | 18 | 35 | 5 | 0 | 0 | 1.25 |
| Zhenglan | 4 | 3 | 7 | 2 | 6 | 6.25 |
| Taipusi | 7 | 15 | 15 | 5 | 8 | 4.80 |
| Xianghuang | 1 | 3 | 13 | 12 | 18 | 8.86 |
| Siziwang | 6 | 30 | 5 | 0 | 0 | 1.59 |
| Damao | 0 | 0 | 10 | 7 | 22 | 10.83 |

**Table A3.** The number and proportion of herders under different grazing rates.

| Steppe Type | <0.4 | 0.4–0.8 | 0.8–1.2 | 1.2–1.6 | ≥1.6 |
|---|---|---|---|---|---|
| Total | 131 (24.21%) | 103 (19.04%) | 82 (15.16%) | 60 (11.09%) | 165 (30.50%) |
| Meadow steppe | 76 (49.67%) | 36 (23.53%) | 15 (9.80%) | 12 (7.84%) | 14 (9.15%) |
| Typical steppe | 36 (11.69%) | 52 (16.88%) | 57 (18.51%) | 34 (11.04%) | 129 (41.88%) |
| Desert steppe | 19 (23.75%) | 15 (18.75%) | 10 (12.50%) | 14 (17.05%) | 22 (27.50%) |

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
