# Peer review of "Impacts of Livestock Production on Grassland Grasshopper Disasters"

_agronomy, doi:10.3390/agronomy14040820_

Round 1

Reviewer 1 Report

Comments and Suggestions for Authors

Dear colleagues,

Thank you very much for interesting studies. I believe your ideas are very important. However, there are a lot of problems with understanding of situation with locusts and acridid outbreaks.

(1) I recommend you to use 'grasshopper or acridid' instead of 'locust' in the title and in many sentences of your manuscript, because actually there are no true locusts in your area (except, I believe, the migratory locust, but this species is commonly associated with reed beds). Sometimes we call the species Oedaleus decorus asiaticus the Mongolian locust, but, in any case, this species is not the true locust. Generally speaking, the locusts are the acridids with presence of the two bioecological forms: gregarious and solitarious with strong differences in behavior, physiology and (often) morphology. This means you try to discuss some results and trends associated with the locusts, but some problems arise, because you should compare your original results, other data for grasshoppers, and data for locusts in very accurate manner.

(2) Page 2 and some other parts — "In addition, the bare land caused by locust outbreaks is a favourable environment for locust spawning and reproduction, leading to recurrent, large-scale grassland locust outbreaks (Gall et al., 2019)." and several similar sentences (e.g. in the next paragraph)— this statement is correct for some species only, because several true locust (e.g. the migratory locust) and many grasshoppers prefer dense vegetation.

(3) Actually you should check and include in your discussions many publications associated either with acridid ecology in Inner Mongolia or with relationships between acridid outbreaks and overgrazing. For instance:

Kang L. Chinese biodiversity 1994

Kang L., Chen Y. Entomologia Sinica 1995

Lockwood et al. Journal of Orthoptera Research 1994

The grassland insects of Inner Mongolia 1991

+ the Special Issue of Journal of Orthoptera Research (2018, 27(1)) with important articles concerning relationships between orthopteran insects and grazing activities.

(4) Introduction — please, characterize goals of your studies in very explicit manner.

(5) Section 2 — in the first paragraph there is some discordance, because you try to declare "reduce ... uniformity" and "reduce ... variety"  

(6) There is some very serious problem with people, because, generally speaking, many (not only general public, but some specialists from the plant protection services!) guess that locusts are some huge insects that can fly. For instance, I could see the specimens of the bush-cricket Tettigonia viridissima determined as the migratory locust. The question is do you sure that all persons involved in your studies told about grasshoppers per se? Please, try to explain this part of your studies.

(7) In some parts of your text, you cite significant levels. Please, avoid these citations, because (1) these data are in the Table 3  and (2) you can use these levels if formulate and discuss null hypotheses.

(8) I am sure that the Conclusion simply repeats the Discussion and may be removed.

(9) Please, edit References according the Journal rules.

Page 1 — Please, insert your names

range lands > rangelands

losses to > losses of

"The damage to grassland caused by the locust plague is devastating, and it is difficult to recover in 20 years, even irreversible, and the economic losses caused are also continuous. The ecological losses caused by locust infestation, such as desertification and degradation of grassland, are incalculable." — please, insert references

Page 2

"the outbreak of Mongolian locusts and Asian locusts by reducing the plant nitrogen content" >

"the outbreaks of the Mongolian (Oedaleus decorus asiaticus) and Asian (Locusta migratoria migratoria) locusts by reducing the plant nitrogen content"

"mining" — please, explain how mining may affect...

Figure 1

degeneration > degradation

Page 3

It is a temperate > There is a temperate

no sense in digits after the points

Latin names of species and genera — in all cases — in italics

Bryodema holdereri > Bryodemella holdereri

last paragraph — please, insert references

Page 4

Interviewees were interviewed — please, re-write

Subsection 3.3

what is α?

Page 6

"This paper aims to study the effect of herders’ production behavior on the frequency of locust disasters in Inner Mongolia." — you should place such sentence in the Introduction.

Page 7

shew > shows or showed (?)

Page 8

damage is Damao > damage is in Damao

Page 11

health of grassland ecology — ecology is a science!

what is "policy intensity"?

Page 12

The biophysical and ecological basis may lie in two aspects. — please, re-write (avoid 'lie')

Comments on the Quality of English Language

The quality of English is moderate. I tried to suggest some improvements, but in some case only. Please, check your text again, may be ask some colleagues to help you.

Reviewer 2 Report

Comments and Suggestions for Authors

Review of MS – Agronomy-2899932 “Impacts of Livestock Production on Grassland Locust Disasters”

General

The authors submit a MS for review, which is of some interest to the community concerned with locust attacks in large areas of Inner Mongolia. Although I find some interesting ideas in the overall data presented by the MS and also confirmations on what is already known, several considerations are necessary which highlight various weaknesses of the MS which make it more of a good technical note necessary to plan general intervention guidelines to contain the locust problem in the area.

In particular, I believe the information provided on the areas investigated is insufficient since for the different areas it would be necessary to specify which species of locust most affect the pastures.

The possible conditions that can favor the proliferation of locusts are different and not specified in the text. Important factors to consider are temperatures, type of soil, humidity, etc.

Throughout the text, verify the name of the locust species distributed in the study area and report the names of the species in italics. Furthermore, report more details if the species indicated are effective pests in the areas investigated.

The question scheme does not highlight whether some herders adopt chemical, biological or other intervention plans to mitigate the effects of these pests.

It is necessary to update the bibliography on the studies done for these areas, for example in the paper "doi: 10.3389/fevo.2021.703220" these effects are highlighted by type of area and in particular greater damage in the pastures of the steppe areas;

Report for the areas the incidence of livestock known from the reports and highlight the origin of the data also with reports already present which highlight the degradation of the pasture;

I don't understand the need to report the formula of the generalized Poisson distribution model in the text.

Round 2

Reviewer 1 Report

Comments and Suggestions for Authors

Dear authors, I guess a few improvements can be made:

p. 2 Take the outbreak of the Mongolian (Oedaleus decorus asiaticus) and Asian (Locusta migratoria migratoria) outbreaks > Take the outbreak of the Mongolian (Oedaleus decorus asiaticus) and Asian (Locusta migratoria migratoria)  locust outbreaks

p.  4 Elymus sibiricus - in italics

Comments on the Quality of English Language

Some minor editing should be done

Reviewer 2 Report

Comments and Suggestions for Authors

Dear Authors,

You submitted the revised version of the MS. The improvements made have increased the overall quality of the MS so in my opinion I believe that the MS is now publishable in harmony with what was found by the other reviewers and the publisher. In the cover letter, however, the you do not respond and refers to a question of mine in particular: "I do not understand the need to report the formula of the generalized Poisson distribution model in the text of the MS.
